# Deep Groundwater as an Alternative Source of Water in the Ogaden Jesoma Sandstone Aquifers of Somali Region, Ethiopia

**Samuel Godfrey [1],\*, Getachew Hailemichael [2] and Charles Serele [3]**

[1]  Water Supply and Sanitation Regional Advisor, UNICEF East and Southern Africa, United Nations Avenue, Nairobi 0100, Kenya
[2]  UNICEF Water Supply and Sanitation Section, UNECA Compound, Zambezi Building, Addis Ababa 1169, Ethiopia
[3]  School of Health and Social Care, University of Essex, Colchester Campus, Colchester CO4 3SQ, UK
\*  Correspondence: sgodfrey@unicef.org; Tel.: +254-741-543-114

**Abstract:** Between 2015 and 2018, the Horn of Africa was affected by a series of climatic-induced events, namely El Nino, La Nina, and the Indian Ocean Dipole. These events modified the variability of rainfall patterns and resulted in extended periods of low rainfall, low recharge, and high evapotranspiration. That situation prompted humanitarian water professionals to finance the transportation of water from selected locations with high groundwater potential through water trucks to areas facing groundwater depletion and drought. To mitigate this, UNICEF identified alternative water supplies by exploring sustainable deeper groundwater sources. This paper describes a three-phase methodology of deep groundwater development of wells in the Ogaden Jesoma sandstone aquifers of the Somali region of the Horn of Africa, to a depth of 600 m below ground level. The methodology included the development of groundwater suitability maps using geological and remote sensing data, hydrogeological ground truthing of the maps, and then test drilling at the selected locations. The results concluded that the deep sandstone aquifer of Jesoma can provide fresh water with yields of 15 L/s to the local population of the Somali region. The study provided insights into deep groundwater identification and development as well as adaptive deep borehole drilling as a source for climate-resilient water supplies.

**Keywords:** groundwater; deep wells; deep aquifers; drought; Somali; Ethiopia

## 1. Introduction

The International Association of Hydrogeologists (IAH) estimates that global withdrawals of groundwater surpassed 900 km$^3$/year in 2010, providing more than 36% of potable water supply, 42% of water for irrigated agriculture, and 24% of direct industrial water supply [1]. Groundwater is now widely recognized as the most extracted raw material in the world with global annual withdrawal rates between of 800–1500 km$^3$/year [2,3]. Doll [4] states that 35% of the water consumed worldwide (4300 km$^3$/year during 1998–2002) is groundwater. Due to the overreliance on groundwater, there are conflicting demands for multiple uses and for achieving multiple Sustainable Development Goals (SDGs) (6,3,11,12 and 15) [5]. The centrality of water to achieve the SDG targets is well acknowledged with groundwater being the primary water source.

In East and Southern Africa, 70% of the population is reliant on shallow groundwater as the primary drinking water source [6]. This is far higher than the global average which notes that 2.8 billion people (or approximately 40% of the world's population) are receiving their water supply from shallow groundwater [7]. In East and Southern Africa, the primary form of extraction of groundwater is through

boreholes fitted with handpumps. Recent reviews of the level of functionality of handpumps indicate an increase in non-functionality of the technology due to, in part, the lowering groundwater levels which are occurring because of reduced precipitation and groundwater recharge [8]. According to the IPCC WGII-AR5, "Climate change will amplify existing stress on water availability in Africa." This is particularly applicable in many African countries where extreme precipitation changes have been aggravating the frequency of droughts and floods during the last 30 to 60 years. A further continued warming in the Indian-Pacific warm pool has also been shown to contribute to more frequent East African coastal flooding and inland droughts over the past 30 years during the spring and summer seasons [9].

Given that groundwater is the primary source of drinking water for the majority of Africa, it is of paramount importance to understand how to identify and develop the "untapped" groundwater in a sustainable and cost-effective manner [10]. Earlier work in the Turkana region of Kenya by UNESCO identified the possible use of a technique termed WATEX (short for water exploration) for deep groundwater development. This technique was evaluated by the International Groundwater Resources Assessment Centre (IGRAC), which concluded that a lack of available information on the methodology was a limiting factor in supporting the studies' conclusions [11]. In Ethiopia, the WATEX approach was piloted in 2011 and modified by UNICEF using open source methods that are outlined in a three-phased methodology used in the Afar and Somali regions. The methodology uses a combination of remote sensing data combined with ground truthing and appropriate deep groundwater drilling methods. This paper builds on an earlier [8] publication and provides a detailed case study of a deep well identified and drilled within the Jesoma sandstone formation of the Somali region in Ethiopia. This paper argues that development of deeper aquifers that are interconnected with surface and groundwater recharge zones is an effective solution in areas of Ethiopia affected by drought.

The data collected to support this argument was undertaken in the drought-affected Gashamo district located in the Somali region of Eastern Ethiopia bordering Somaliland (Figure 1). Gashamo is an area in the Ogaden Sedimentary Basin located in the southeastern part of Ethiopia that covers about 350,000 km$^2$ and has a population of 145,000. The population of Gashamo is reliant on collecting rainwater from birkas or hafir dams, which is only possible during short rains. As a result, communities migrate to other areas or rely on water being tanked by the government and UNICEF from more than 40 km away.

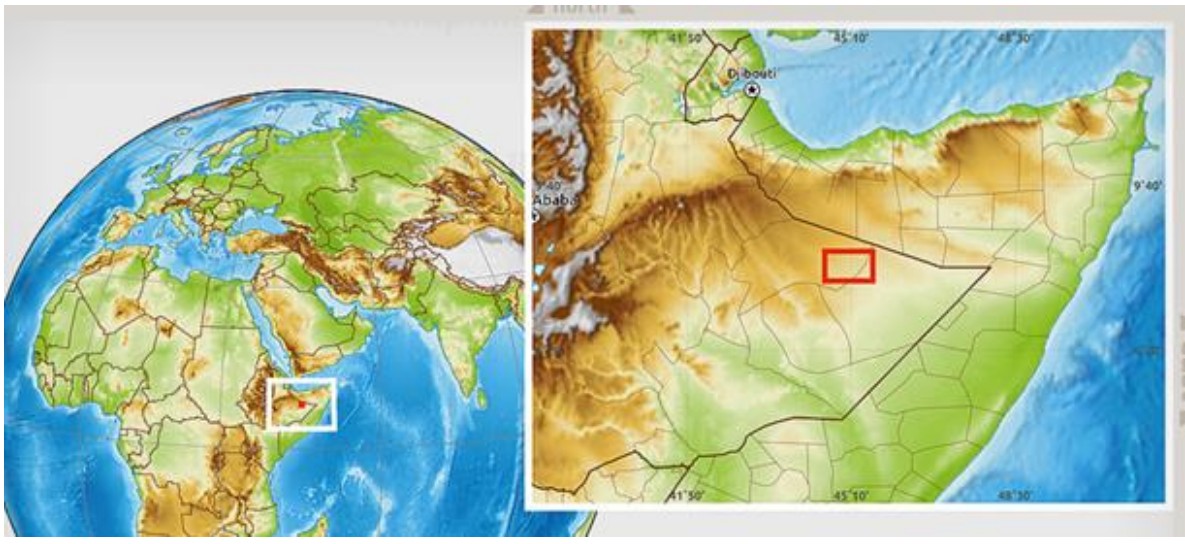

**Figure 1.** Location of the study region. Source maphill access 13th August 2019.

## 2. Materials and Methods

The project followed a three-phased approach to identify the most promising site for drilling. Remote sensing, ground truthing, and geophysical surveys enabled determination of well drilling sites in the Jessoma sandstone aquifer of the Gashamo district. Due to the limited number of test wells and the lack of deep groundwater investigation data in the study area, UNICEF initiated a deep groundwater survey in the area using the three-phase methodology outlined in [8] and in Figure 2.

| PHASES OF IMPLEMENTATION | | |
|---|---|---|
| **PHASE 1:**<br><br>**Groundwater Potential Map (1:250,000) from remote sensing data** | PHASE 2:<br><br>Geophysical and hydrogeological map for drilling site identification (1:100,000) | PHASE 3:<br><br>Test drilling |
| **2016** | 2017 | 2018 |

**Figure 2.** Thematic layers.

The methodology consists of an overlay analysis of different groundwater-relevant information (derived from remote sensing products, existing maps and report, and geophysical survey) to create a groundwater potential map that allows identifying areas of high probability of groundwater occurrence for drilling of productive boreholes. The remote sensing-derived thematic layers were obtained through a collaboration with the European Union Joint Research Centre (EU-JRC). It is acknowledged that recharge of an aquifer refers to the infiltration of water from the unsaturated zone to the saturated zone is dependent upon several factors, such as topography, geological structures and drainage network. [12]. Because these factors are interdependent, considering a single factor to explain the recharge process reduces the reliability of the estimates for a given region. Therefore, the adopted methodology is based on expert judgement overlay of four aquifer recharge-related thematic layers, mainly geomorphology, drainage density, permeability of geological formations, and lineament density (structural density). A weighting system was used to analyze the thematic layers and is outlined in Table 1. However, as each thematic layer influences differently the potential aquifer recharge, a weighting process was applied to consider the importance of each factor in relation to the others [13]. Hence, the more a factor influences groundwater occurrence, the greater its relative importance resulting in a high weight [14].

The weightage was validated with field hydrogeological and geophysical surveys. Specific spots were identified for further geophysical survey using vertical electrical sounding and profiling to understand the underground lithology and water-bearing formations. The resultant groundwater potential map was then used to identify suitable locations for drilling based on access for the local population, security, and access for the drilling contractor. Due to the geologic nature of the area, the lithologic formation indicated an alteration between soft (collapsible), soft (not collapsible), medium, slightly hard, and hard formations. Reverse circulation methods had been employed on other wells in the area using tri-cone bits designed for soft formations, but these had a slow rate of penetration even when applied with bentonite clay to avoid collapse. The method used in this study involved reverse circulation with water stabilization. A required amount of more than 20,000 L of water per day was needed to stabilize the well during drilling. All data collected from the three-phased methodology were used to develop the standard terms of reference, estimated bill of quantities, and technical specification for future groundwater drilling engineering contracts in the Somali region.

**Table 1.** Weighting for layers.

| Layer | Parameter | Factor Weight | Class and Class Weight | | | |
|---|---|---|---|---|---|---|
| | | | 1 | 2 | 3 | 4 |
| 1 | Geomorphology | 0.3 | Flat low- lying grabens | Gentle to flat horst | Gentle to steep horst | Steep and mountainous |
| | | | 50 | 30 | 15 | 5 |
| 2 | Drainage/drainage density | 0.25 | High | Moderate | | |
| | | | 70 | 30 | | |
| 3 | Hydrogeology/permeability | 0.23 | High | Low to moderate | Aquiclude | |
| | | | 60 | 35 | 5 | |
| 4 | Structure/structural density | 0.22 | Major (regional) | Moderate | Local | Poor |
| | | | 55 | 25 | 15 | 5 |
| | Total | 1 | 58.4 | 30.05 | 8.95 | 2.6 |
| | Classification | | High | Moderate | Low to moderate | Low |

## 3. Results

### 3.1. Geological and Hydrogeological Conceptual Model

The lithologic units identified in the Gashamo area consist of Tertiary Mesozoic sedimentary rocks with intercalation of sandstones, limestones, and gypsum, which, in general, is named "Jesoma sandstone." The basin is underlain by very thick succession of Late Paleozoic, Mesozoic, and Cenozoic (Tertiary) sedimentary rocks. It is characterized by flat to undulating plain geomorphology, where the surface geology is Mesozoic sedimentary deposit with a thickness that varies from less than 200 m to more than 1000 m below ground. The identified formation consists of gray and multicolored sandstone intercalated with multicolored shale and lateritic layers. Tertiary Jesoma sandstone covers almost all the Ogaden Basin, including the project area, and is composed of poorly sorted and poorly consolidated sandstone. Two test wells had been drilled in this formation in 2015 and 2016, namely in Harshin and Yaole locations [15]. These well logs showed that the Jesoma sandstone thickness is variable and can range from depths of 130 m below ground. Data from these wells indicated that the water quality in the upper layers from the test wells is brackish.

Using the overlay methods outlined in [8], a conceptual model was developed and outlined in Figure 2 below. The conceptual model identified an absence of a retaining layer at shallow depth due to the high transmissivity of the sandstone formations. Recharge was noted to percolate deeper until it intercepts the underlying limestone or clay layers of the upper cretaceous formation. As per the interpretation outlined in Figures 3 and 4, the water-bearing layers are expected to be located between depths of 300 m to 450 m below ground level.

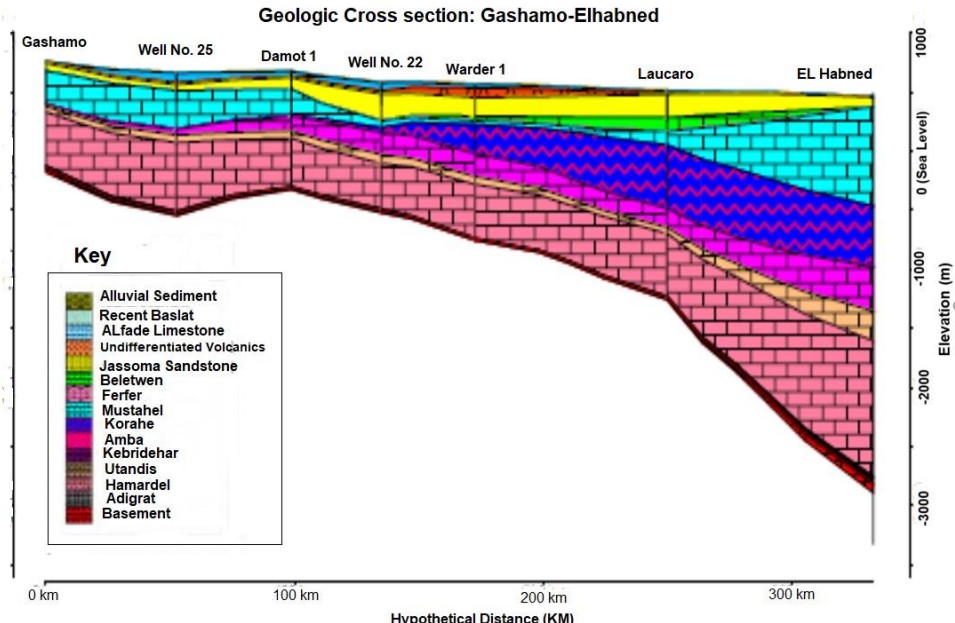

**Figure 3.** Geology of Gashamo.

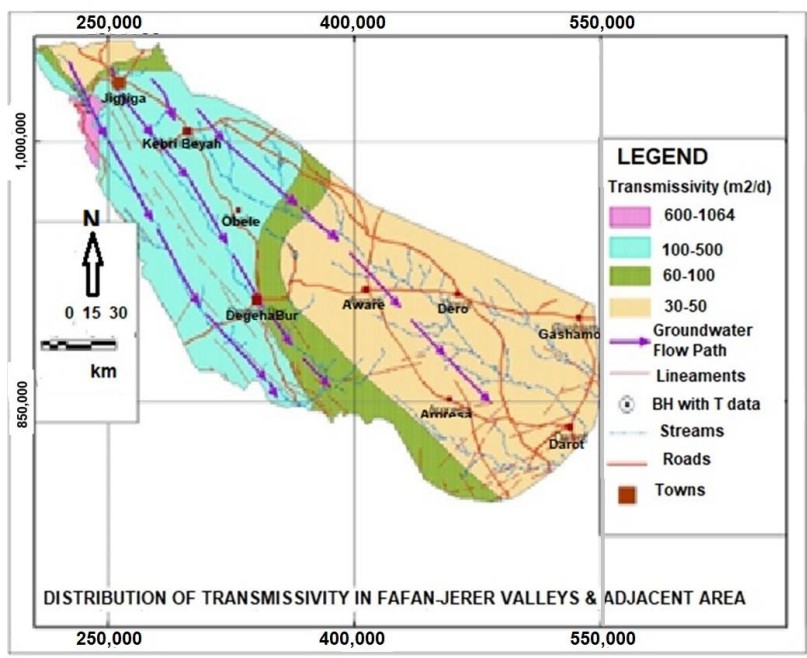

**Figure 4.** Hydrogeological conceptual model for Gashamo.

## 3.2. Geophysical Survey

The second phase of the methodology included confirmation of the hydrogeological model using field level Vertical Electrical Soundings (VES). Six VES tests were conducted along the west to east direction. The VES interpretation is outlined in Table 2.

The results of the VES indicated that groundwater generally occurs at depths below 300 m in the Auradu limestone and 450 m to 500 m in the Jessoma sandstone. In the Urandeb formation the quality of water becomes more saline with increasing depth. Even at shallower depth, from 0 to 35 m, the groundwater quality is highly deteriorated. Based on the combination of the conceptual model and the VES readings, the study observed that the Gashamo site is composed of highly permeable tertiary sedimentary formations dominantly composed of poorly sorted and poorly consolidated sandstone.

The regional groundwater flow in the area is from northwest to southeast, which could most probably be controlled by the piezometric level of the Indian Ocean. The groundwater in the area is deep, mainly caused by high permeability of the unconsolidated sedimentary–sandstone (main aquifer) and the groundwater flow that is discharged to the Indian Ocean and controls. This is certified by the available information from the available test borehole data where the static water level (SWL) is 324 m below the ground surface, the average regional hydraulic gradient is calculated to be 0.000941.

**Table 2.** Estimated water level depth from Vertical Electrical Soundings survey.

| No. | NAME | Estimated SWL (meter) | | Aquifer Resistivity (Ωm) | Remark |
|---|---|---|---|---|---|
| 1 | VES-1 | 292.7 | 9.4 | Kayder labile (site)—Probably fresh | |
| 2 | VES-2 | 287.8 | 6.4 | Probably brackish | |
| 3 | VES-3 | 349.1 | 3.7 | Anamadobe (site)—Brackish | |
| 4 | VES-4 | 360.4 | 12.1 | Most probably fresh | |
| 5 | VES-5 | 358.0 | 22.7 | Fresh | |
| 6 | VES-6 (calibration) | 314.3 | 14.8 | Fresh, as per Balijano well water quality data EC = 1310 mS/cm | |

The groundwater level depth along the transect (Gashamo area) is estimated based on the hydraulic gradient and vertical electrical sounding interpretation. The conceptual hydrogeological x-section indicates a high probability of deep water that may be brackish due to the gypsum in the aquifer. For drilling purpose, VES site 5 was selected as the specific site for well siting at a depth of 600 m. Resistivity readings were taken during the drilling to interpolate the electrical conductivity of the well and its potential to produce saline or brackish water. The results from this study are outlined in Figure 5 and showed a gradual increase of resistivity, meaning the presence of fresh water with lower conductivity.

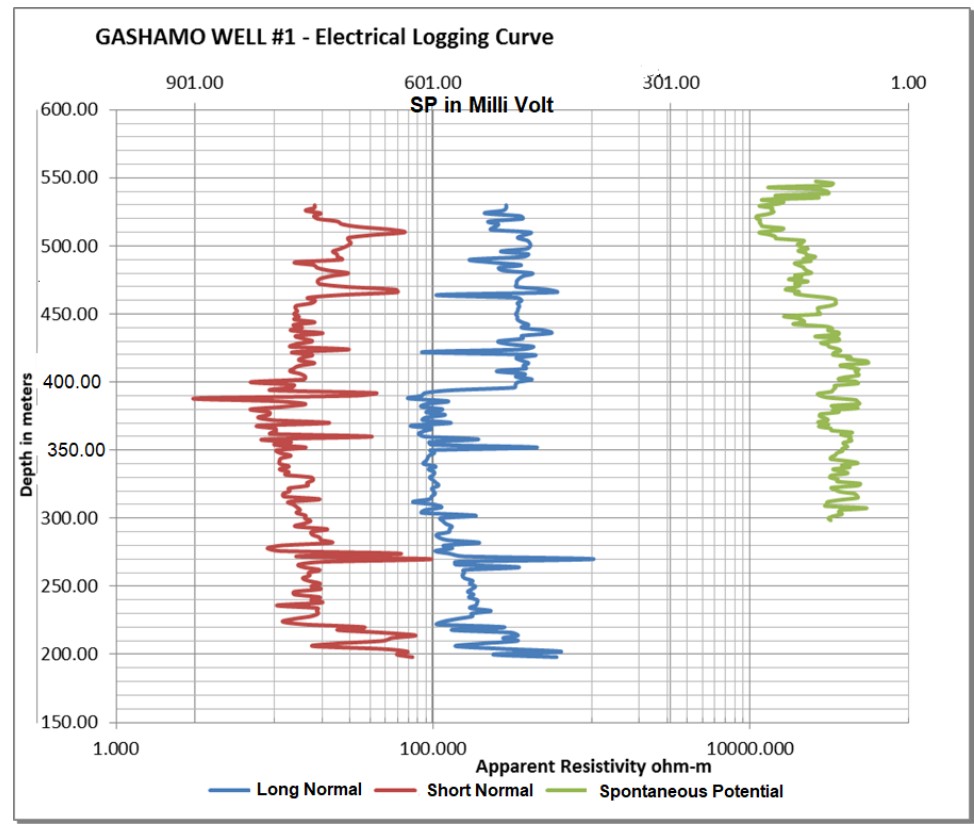

**Figure 5.** Electrical logging of well 1.

### 3.3. Test Borehole Drilling

Based on the conceptual model of the lithologic formation and the geophysical survey, the study concluded that the most climate-resilient water supply solution in the Gashamo district is deep drilling of groundwater from the sandstone/limestone aquifer. As predicted by the hydrogeological conceptual model and the geophysical study, the well drilled at 600 m depth was productive and provides fresh and non-brackish water. The static water level was estimated from the rims of water marks on the drill pipes, the interpretation of the electrical logging, and a change in water temperature from the surface temperature of 42 °C. A constant-rate pumping test of thirty-six hours was undertaken using a Shakti 35 HP pump motor coupled with a Lowara Pump Head, SANS 1973-8(MTA) Panel Board, Lister, 93.5KVA Generating set. The hot water (at least 50 °C) pump was installed at a position of 423 m and the discharge from the well was measured from 4.5 L per second. Medium-class stainless steel riser pipes of 63 mm were used for test pumping, due to the temperature of the water, using specific design of the thread shape and depth to withstand the load and vibration of the pump and the riser pipe itself. The pipes were also further welded to make sure they did not disconnect in the well. The pumping test concluded that the static and dynamic water levels were found to be, respectively, 381 and 385.21 m below ground and indicated a drawdown of 4.21 m, as outlined in Figure 6.

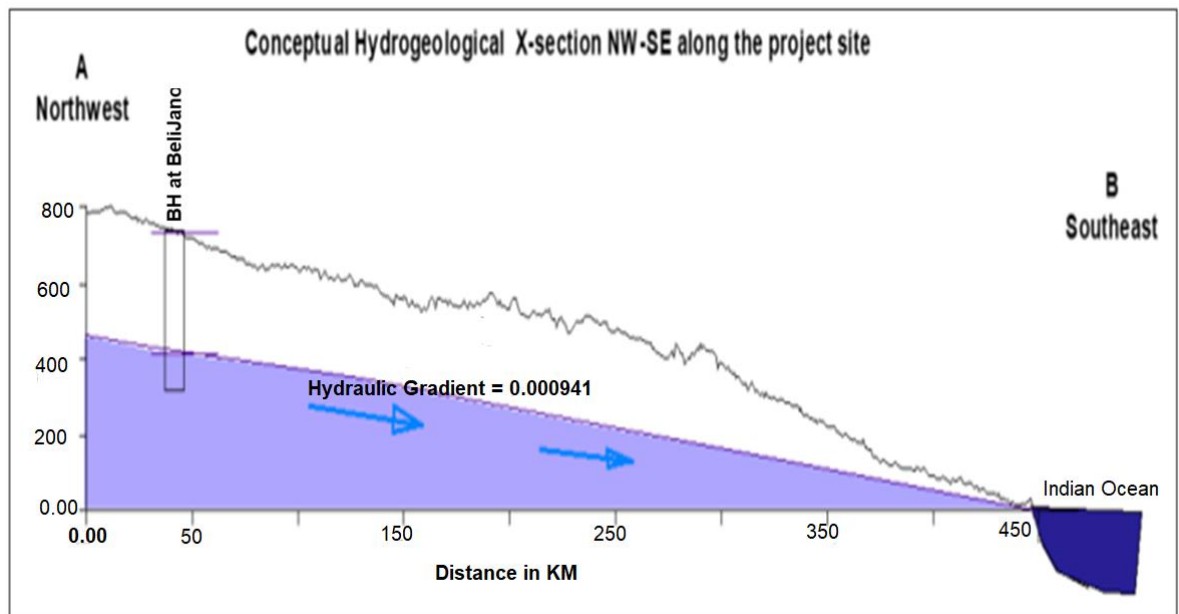

**Figure 6.** Conceptual model of the lithologic formation.

After the constant-rate pumping test of 36 h (Figure 7), the pump is shut off and a recovery test was conducted. The water level stabilized after only sixty minutes to 381.02 m, which is only 0.2 m lower than the static water level. This rapid recovery time and low decrease in the water level is synonymous with a high transmissivity aquifer. Based on the pumping test, an estimated pumping yield of 15 L/second was determined for this deep well (Figure 8).

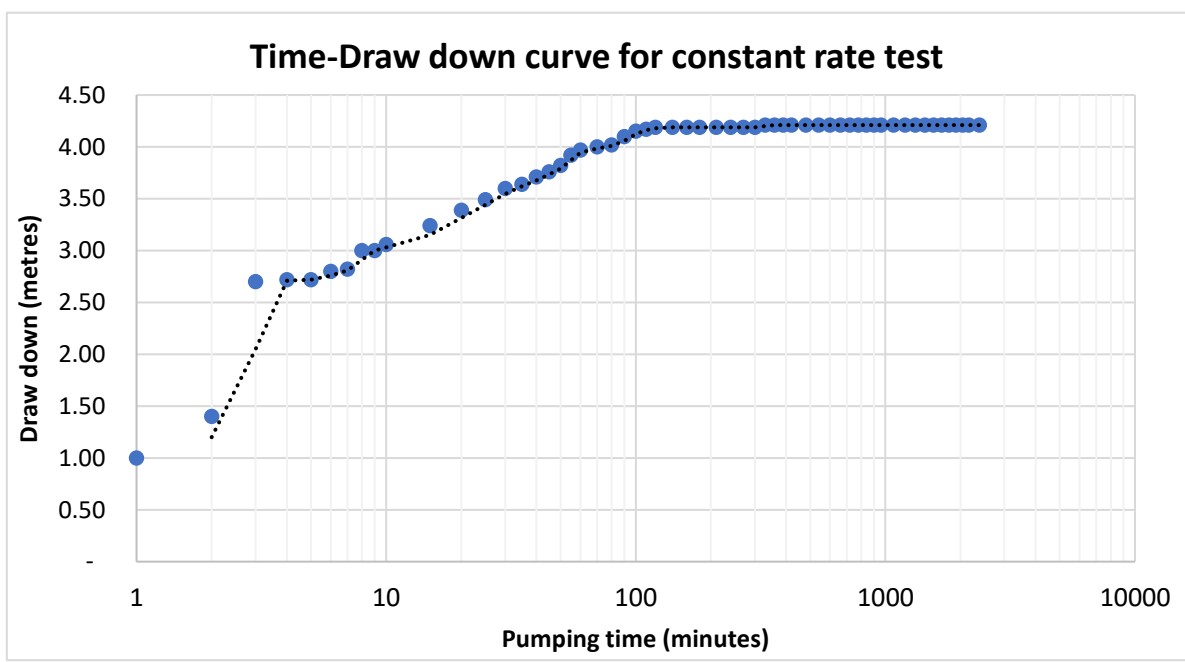

**Figure 7.** Constant-rate test drawdown curve.

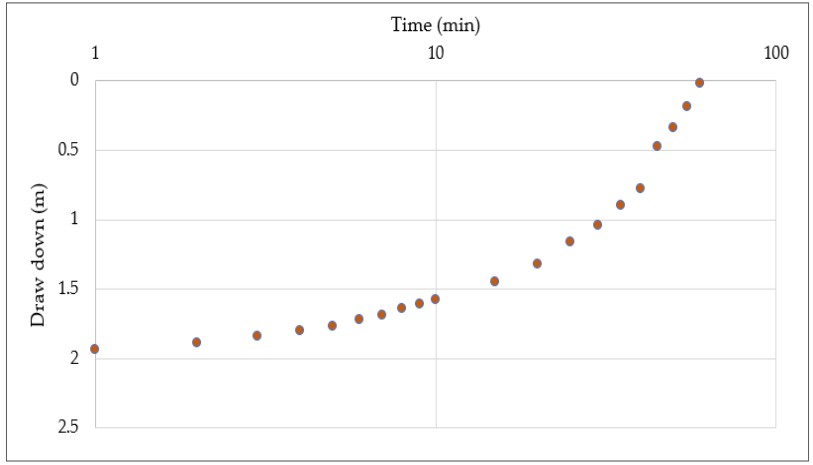

**Figure 8.** Recovery test drawdown curve.

## 4. Discussion

This paper complements the results of previous drilling studies undertaken in Eastern Ethiopia in shallow aquifers. It argues that from an economic perspective it is value for money (VfM) to invest in deeper aquifer exploration and from an environmental perspective it is not damaging to the broader ecosystem.

Economically, this paper argues that an initial capital expenditure (CAPEX) investment in deeper aquifers provides a more resilient water source. During the El Nino drought, water engineers and water sector planners explored the use of on-plot desalinization plants to extract brackish shallow groundwater, treat, and pump it for domestic and agricultural use. The energy demand to ensure efficient operation of the plants, combined with high levels of calcification, precluded the use of solar water pumping and solar energy, and resulted in the installation of diesel-generator systems. A value for money (VfM) analysis revealed that the operation cost of the generators was potentially harmful for the environment and resulted in high water tariffs which made the water unaffordable for domestic use.

Environmentally, this paper supports earlier studies from the Elidar district of the Afar region of Ethiopia, which showed that shallow wells located in alluvial deposits (Dobi 1 and 2) produced

brackish and saline water [8]. The Elidar wells were noted to not only have challenges in water quality but also of water quantity (3 L/s). The challenges in the water quantity and quality could be explained by the fact that the region of Elidar experienced low rainfall and elevated temperatures, and due to high evapotranspiration, and that the shallow aquifers are at risk of salinization. However, it also demonstrates the vulnerability of the shallow aquifers to surface-groundwater interactions, which results in both geochemical contamination and potential microbial contamination from surface runoff in areas with limited sanitation. Furthermore, isotope analysis of shallow aquifers in Ethiopia support earlier studies by [16] in the Gidabo basin of Ethiopia, which note the complex nature of the aquifer systems and distinct difference between the shallow and deep groundwater flows. They used $\delta^{18}O$ and $\delta^2H$ values to estimate the age of the groundwater and they concluded that there is a direct relationship between the deep groundwater flows and rainfall patterns. Therefore, a systematic understanding of deep lithology was undertaken using appropriate hydrogeological investigations. The conceptual model of lithological formation and the geophysical survey concluded that the most climate-resilient water supply solution in the Gashamo district is to drill in the deep sandstone/limestone aquifer.

Deep drilling results have shown that the Jesoma sandstone formation can provide fresh water with yields of 15 L/s. The presence of quaternary deposits makes this Jesoma sandstone formation an exploitable high-potential aquifer, especially in its deep part, which is characterized by cherty limestone limited by impermeable layers. The exploitation of this deep aquifer of sandstone can provide water to the local population and improve the equitable access to safe and clean drinking water in the drought-affected areas of the Somali region. However, appropriate integrated water resources management is required to ensure that deeper boreholes are not overpumped and do not result in either surface-to-groundwater contamination or overextraction.

Constraints in deep aquifer exploitation include the lack of deep well drilling capacity within the Ethiopian drilling sector. A lack of experienced deep drilling capacity could result in the loss of drilling tools, finance, and time. Lessons learned indicated the need to invest in reverse circulation drilling to avoid well collapse.

## 5. Conclusions

The study supports the three-phased approach and emphasizes the need for a systematic understanding of the lithology with appropriate hydrogeological investigation, predrilling arrangement, contract management activities, comprehensive drilling supervision, backup support from project management, and strict project monitoring activities. The high CAPEX cost is necessary to successfully complete the drilling in a place where water is scarce. The study concludes that the Jesoma sandstone formation can provide fresh water with yields of 15 L/s. The presence of quaternary deposits makes the Jesoma sandstone formation an exploitable high-potential aquifer, especially in its deep part, which is characterized by cherty limestone limited by impermeable layers. This paper contributes to improve the equitable access to safe and clean drinking water for the local population of the drought-affected Somali region and to the broader Somaliland region. The paper provides a clear justification for investment in "resilient" water solutions that provide water throughout the year without requiring supplementary emergency water trucking.

**Author Contributions:** Conceptualization, S.G. and G.H.; Methodology, S.G. and G.H.; Formal analysis, S.G., G.H. and C.S.; Writing, original draft preparation, S.G., G.H.; Writing, review and editing, S.G., G.H. and C.S.; Visualization, S.G., G.H. and C.S.; Supervision, S.G.

**Funding:** This research was funded through the European Union (EU) RESET program in Ethiopia and from the Government of Japan.

**Conflicts of Interest:** The authors declare no conflict of interest.

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
