# Peer review of "Deep Groundwater as an Alternative Source of Water in the Ogaden Jesoma Sandstone Aquifers of Somali Region, Ethiopia"

_water, doi:10.3390/w11081735_

Round 1
Reviewer 1 Report
The aim of the paper and its scope is very interesting and valuable. After correcting and supplementing it deserves to be published.
1. The title and introduction refer to climate change. This is an important issue, climate change affects water scarcity. further part of the research refers to the designation of alternative access to water to protect against the effects of these changes. I suggest changing the title to: “The use of deep groundwater as an alternative source of water in ...”
2. Line 84, Figure 2. The map of ethnic groups is not related to the topic. On the Map Location (..) there is no description of which administrative units have been included.
3. Line 128. On the basis of Figure 2 you can’t tell what depth there are groundwater.
4. Line 131. Figure 3. The scale in numeric format is not needed. It makes it difficult to read the map.
5. Add a diagram of the thematic layers which were used.
6. What other places have been designated by experts? Explain why it was decided to drill in the selected location.
7. What weight has been given to the thematic layers.
8. Based on Figure 2, it is not possible to determine at what depth groundwater is.
9. Line 138. Table 1. The symbols X, Y, Z are not explained.
10. In Material and methods, complete the information about VES method (methodology, principle of scientific research, equipment).
11. Why you pumped only on stage 1. In European countries, pumps are pumped at 3 degrees. Please quote the general methodology of testing pumps for Ethiopia.
12. Has the filtration coefficient been calculated for the aquifer based on the trial pumping?
13. Line 195. Which empirical calculations (formulas, assumptions) were used to calculate the well's efficiency.
Author Response
Reviewer 1
1 | The title and introduction refer to climate change. This is an important issue, climate change affects water scarcity. further part of the research refers to the designation of alternative access to water to protect against the effects of these changes. I suggest changing the title to: “The use of deep groundwater as an alternative source of water in ...” | Title has been changed as suggested |
2 | Line 84, Figure 2. The map of ethnic groups is not related to the topic. On the Map Location (..) there is no description of which administrative units have been included. | Ethnic group map has been removed |
3 | Line 128. On the basis of Figure 2 you can’t tell what depth there are groundwater. | The text has been changed a new figure 2 added. |
4 | Line 131. Figure 3. The scale in numeric format is not needed. It makes it difficult to read the map. | The figure has been edited as necessary. |
5 | Add a diagram of the thematic layers which were used. | Added |
6 | What other places have been designated by experts? Explain why it was decided to drill in the selected location. | Some text has been added |
7 | What weight has been given to the thematic layers. | Weightage has been added |
8 | Based on Figure 2, it is not possible to determine at what depth groundwater is. | This has been resolved with the addition of a new figure |
9 | Line 138. Table 1. The symbols X, Y, Z are not explained. | These have been deleted |
10 | In Material and methods, complete the information about VES method (methodology, principle of scientific research, equipment). | This point is not clear |
11 | Why you pumped only on stage 1. In European countries, pumps are pumped at 3 degrees. Please quote the general methodology of testing pumps for Ethiopia. | In Africa, a single test pumping is followed. I have added some text. |
12 | Has the filtration coefficient been calculated for the aquifer based on the trial pumping? | Not included |
13 | Line 195. Which empirical calculations (formulas, assumptions) were used to calculate the well's efficiency. | Revised |
Reviewer 2 Report
General comments
remote sensing such as GRACE are not used or highlighte din the method
also scaling down of the remote sennsing to site specific (groud truthing) is not explicitly mentioned
The none-uniquness problem solution of the inverse method (geophsyics) solution is not mentioned.
Resilence and climate change are mentioned but with out much detail to the study area in terms of water resource. It needs to be assessed or left out/remove.
Methodology
The three phased approach needs to be presented in a flow chart and how those relate to each other via feed back/loop etc. It lacks clarity and the flow of the idea gets in disarray.
The hydrogeological conceptual model lacks quanitative resource /reserve estimate.Moreover, hydraulic parameter (storage) is not explictly assessed and employed for resource estimate. Multi-layer aquifer assessment (shallow vs deep aquifers) is not addressed via inter leakage/upward/diownward gradient etc, despite a huge meaning interms of climate change and resilence. T this can be done using
Through flow method
Hydrograph fluctuation method ( Area, Storage coefficent, seasonal hydraulic fluctuation)
groundwater balance method ( please see sugested references below)
Technical words suich as:
initial CAPEX (Capital Expenditure) investment and
Value For Money (VfM) are shown up in the discussion with out details in the proceeding sections.
Minor comments
The article requires editing. To mention few:
line 89 ( needs space between "andthe")
Figure 2( Legend- Transmissivity - "m2/day". "2" should be superscript
Few references below-to address the coments above:
§ Yihdego Y and Webb J.A., 2015 “Use of a conceptual hydrogeological model and a time variant water budget analysis to determine controls on salinity in Lake Burrumbeet in southeast Australia” Environmental Earth Sciences Journal. 73 (4):1587-1600. http://link.springer.com/article/10.1007/s12665-014-3509-x
Yihdego, Y, Khalil, A (2017).
Groundwater Resources Assessment and Impact Analysis Using a Conceptual Water
Balance Model and Time Series Data Analysis: Case of Decision Making Tool. Hydrology, 4(2), 25.
doi:10.3390/hydrology4020025. https://www.mdpi.com/2306-5338/4/2/25
Author Response
Reviewer 2
1 | The three phased approach needs to be presented in a flow chart and how those relate to each other via feed back/loop etc. It lacks clarity and the flow of the idea gets in disarray.
| This is now included |
2 | The hydrogeological conceptual model lacks quanitative resource /reserve estimate.Moreover, hydraulic parameter (storage) is not explictly assessed and employed for resource estimate. Multi-layer aquifer assessment (shallow vs deep aquifers) is not addressed via inter leakage/upward/diownward gradient etc, despite a huge meaning interms of climate change and resilence
| Weightage has been included based on this recommendation |
3 | Technical words suich as: initial CAPEX (Capital Expenditure) investment and Value For Money (VfM) are shown up in the discussion with out details in the proceeding sections.
| These have been explained in the text |
4 | Editing of english | This has been included |
Reviewer 3 Report
Climate resilient deep groundwater investigation and 2 development in the Ogaden Jesoma sandstone 3 aquifers of Somali Region, Ethiopia
This manuscript used the WATEX three-phased approach in an area of Somali Region, Ethiopia. Moreover, it emphasizes the need for a systematic understanding of the lithology with appropriate hydrogeologic investigation before drilling.
The topic is very interesting, especially in arid areas where it is important to drill productive well and to withdraw good quality groundwater.
However, the paper is very difficult to read.
First of all, the Watex method is not described but only a reference is reported. The geological setting is poor, and a geologic map and or cross section is lacking.
The results are not clearly supported by figures and tables.
I think that a more complete and clear description of methods and results must be inserted, before to accept and publish the paper.
ABSTRACT
LINE 31-32: the sentence is probably incomplete. Please, modify it.
INTRODUCTION
In the introduction is not clear what is WATEX technique and if it was used in the study area.
MATERIALS AND METHODS
Line 87-88: the word “that” can be removed
Line 89: andthe : separate and the
The explanation of the method must describe all the single steps. Instead in the paper only
references to the bibliography are reported (using the three-phase methodology outlined in (8)). The reader in this way has great difficulty understanding the method. Please, be more detailed and illustrate the method in a complete way.
line 103: what does it mean “a based-on experts’ judgement”? RESULTS HYDROGEOLOGICAL CONCEPTUAL MODEL It could be useful to add a geologic map of the area, and if possible a cross section. Because of it is very difficult to understand the used method (there only a reference, but not a description) I’m not sure that the Hydrogeological conceptual model must be inserted in the results or in the method / study area. Is the reconstruction of Hydrogeological conceptual model the first step of the method? Otherwise, I moved it in the methods/ study area. Line 122: “well logs showed”: The paper (15) is unpublished, and then the reader is not able to have an idea on the well logs. Please, insert in the paper a figure with well logs. Line 124: what are the reasons of the presence of brackish water? Line 124-125: “Using the overlay methods outlined in (8), a conceptual model was developed and outlined in figure 2 below.” It is not clear how the authors developed the conceptual model. Line 122-130: “The conceptual model identified an absence of a retaining layer at shallow depth due to the high transmissivity of the sandstone formations. Recharge was noted to percolate deeper until it intercepts the underlying limestone or clay layers of the upper cretaceous formation. As per the interpretation outlined in figure 2, the water bearing layers are expected to be located between depths of 300 m to 450m below ground level.” These sentences are not supported by a clear description of the geologic setting. What does it means: “The conceptual model identified an absence of a retaining layer at shallow depth due to the high transmissivity of the sandstone formations”? Why the authors say “the water bearing layers are expected to be located between depths of 300 m to 450m below ground level”? GEOPHYSICAL SURVEY 138: “interpreted water quality condition from VES survey”. 139-140: “The results of the VES indicated that groundwater generally occurs at depths below 300 m depth in the Auradu limestone and 450 m to 500 m depth in the Jessoma sandstone.” Because of the VES are not located on a map, it is impossible to check the information. Please, insert a mp with geographic denomination. Tab. 1 is not clearly described (see also the revision of figures and tables). Aquifer resistivity and remark (water quality???) are connected on the base of what? Do you used a reference? Do you have also indication of EC? Figure 4. Electrical logging of well 1. The figure is not clear. What is SP? What are the three lines? the figure must be described in detailed. Line 157-161: “For drilling purpose, the VES site 5 was selected as the specific site for well siting at a depthof 600 m. Resistivity readings were taken during the drilling to interpolate the electrical conductivity of the well and its potential to produce saline or brackish water. The results from this study are outline in the figure 4 and they showed a gradual increase of resistivity meaning the presence of a fresh water with lower conductivity.” Gashamo well 1 and ves site 5 are the same point? “Resistivity readings were taken during the drilling”. How? “interpolate the electrical conductivity of the well”. Where are these data? FIGURE Figures have a low resolution. Please, improve the resolution Figure 1. Location of the study region. I think that more than the ethnic groups, it could be more useful to have a geological or hydrogeological map. Please, modify it. Figure 2. Hydrogeological conceptual model for Gasham. In legend the authors added BH with T data, but this simbol is not present in the figure. Please, add it. Table 1. please, modify “water le vel depth” in “water level depth”. Please, describe what are X Y Z Estimated SWL What is remark? Is it water quality??? Figure 3. Conceptual model of the lithologic formation. The figure must be improved adding the lithologic formation in the cross section.
Author Response
Reviewer 3
1 | First of all, the Watex method is not described but only a reference is reported. The geological setting is poor, and a geologic map and or cross section is lacking.
| This is a modified version of the Watex and the weighting and the methodology have now been included |
2 | LINE 31-32: the sentence is probably incomplete. Please, modify it.
| Modified |
3 | Line 87-88: the word “that” can be removed. Line 89: andthe : separate and the | Done |
4 | The explanation of the method must describe all the single steps. Instead in the paper only | Done |
5 | It could be useful to add a geologic map of the area, and if possible a cross section. | Done |
6 | Expert judgement | This is explained using the added text and weightage table |
7 | Line 122: “well logs showed”: The paper (15) is unpublished, and then the reader is not able to have an idea on the well logs. Please, insert in the paper a figure with well logs.
| The government of the Somali region are not forthcoming in giving this log. It is therefore not included |
8 | Line 124: what are the reasons of the presence of brackish water?
| Text has been added |
9 | Line 124-125: “Using the overlay methods outlined in (8), a conceptual model was developed and outlined in figure 2 below.”
| Included now |
10 | Line 122-130: “The conceptual model identified an absence of a retaining layer at shallow depth due to the high transmissivity of the sandstone formations. Recharge was noted to percolate deeper until it intercepts the underlying limestone or clay layers of the upper cretaceous formation. As per the interpretation outlined in figure 2, the water bearing layers are expected to be located between depths of 300 m to 450m below ground level.” | Correct |
11 | 138: “interpreted water quality condition from VES survey”.
| Table focusses now on water depth only |
12 | 139-140: “The results of the VES indicated that groundwater generally occurs at depths below 300 m depth in the Auradu limestone and 450 m to 500 m depth in the Jessoma sandstone.” Because of the VES are not located on a map, it is impossible to check the information. Please, insert a mp with geographic denomination.
| The text has been modified to reflect the results. Due to security reasons and access in this volatile Somali region, we can not provide a further map. |
13 | Gashamo well 1 and ves site 5 are the same point? | Yes |
14 | I think that more than the ethnic groups, it could be more useful to have a geological or hydrogeological map | This has been modified |
15 | In legend the authors added BH with T data, but this simbol is not present in the figure. Please, add it. | Done |
16 | Table 1. please, modify “water le vel depth” in “water level depth”. Please, describe what are X Y Z Estimated SWLWhat is remark? Is it water quality???
| The table has been modified |
17 | Figure 3. Conceptual model of the lithologic formation. The figure must be improved adding the lithologic formation in the cross section.
| This has been modified |
Reviewer 4 Report
The study describes a three-phase methodology of climate resilient deep groundwater development of wells in the Ogaden Jesoma sandstone aquifers of the Somali Region, Ethiopia. While the topic is interesting to readers, the current version of the manuscript is poorly organized with several grammatical errors. The writing and contents of the research is very hard to follow. Overall the English is poor, and requires editing from a native English speaker. Based on these points, I recommend major revisions of the manuscript. My specific comments are as follows:
1. Abstract: Too much background information. Description of methodology and results is insufficient. Should clearly and concisely mention the stages of the mentioned three-stage methodology. Overall, abstract requires a judicial rewriting.
2. Page 1, Line 31: Incomplete sentence. Please add the missing words to complete the sentence.
3. Page 2, Line 47: Please revise the sentence " This far higher than the global average which notes that...." for better readability.
4. Page 2, Lines 50-53: The sentence "Recent reviews.......groundwater recharge" requires revising and rewriting.
5. Page 3, Line 86: Are you referring to a specific project?
6. Methodology lacks clarity. So far I understand, the research is based on earlier published documents. Although you gave credit to this published work, the three-phase methodology and the steps are not clearly articulated in the methodology section of the current version of the manuscript. For readability and better understanding of the contents, the methodology followed in this research must be clearly mentioned.
7. Page 4, Lines 116-125: Did you gather this information from your own conceptual model? If yes, you need to provide a description of the conceptual model along with the initial and boundary conditions in the methodology section. If not, these texts should appear in the background and methodology section.
8. Page 5, Line 135: Provide brief description of the Vertical Electrical Soundings (VES) in the methodology section. You don’t need to mention the methodology again in the result section.
9. Page 6, Lines 157-159: Again you are mixing up results and methodologies.
10. Page 6, Lines 159-161: Grammar error (The results from this study are outline). More explanations with interpretation of the results are needed here.
11. Page 6, Lines 171-173: ‘Reverse circulation……to avoid collapse” might go to methodology section.
Author Response
Reviewer 4
1 | Abstract: Too much background information. Description of methodology and results is insufficient. Should clearly and concisely mention the stages of the mentioned three-stage methodology. Overall, abstract requires a judicial rewriting. | The abstract has been modified and edited. |
2 | Page 1, Line 31: Incomplete sentence. Please add the missing words to complete the sentence. | Done and included |
3 | Page 2, Line 47: Please revise the sentence " This far higher than the global average which notes that...." for better readability. | Is has been added to the sentence |
4 | Page 2, Lines 50-53: The sentence "Recent reviews.......groundwater recharge" requires revising and rewriting. | Done |
5 | Page 3, Line 86: Are you referring to a specific project? | Modified |
6 | Methodology lacks clarity. So far I understand, the research is based on earlier published documents. Although you gave credit to this published work, the three-phase methodology and the steps are not clearly articulated in the methodology section of the current version of the manuscript. For readability and better understanding of the contents, the methodology followed in this research must be clearly mentioned. | We have now added some more clarity around the methodology. Some journals prefer to reference other work and we acknowledge the need to improve this section. |
7 | Page 4, Lines 116-125: Did you gather this information from your own conceptual model? If yes, you need to provide a description of the conceptual model along with the initial and boundary conditions in the methodology section. If not, these texts should appear in the background and methodology section. | Yes and now included as recommmended |
8 | Page 5, Line 135: Provide brief description of the Vertical Electrical Soundings (VES) in the methodology section. You don’t need to mention the methodology again in the result section. | Now clear |
9 | Page 6, Lines 157-159: Again you are mixing up results and methodologies.
| Text has been modified |
10 | Page 6, Lines 159-161: Grammar error (The results from this study are outline). More explanations with interpretation of the results are needed here. | Changed |
11 | Page 6, Lines 171-173: ‘Reverse circulation……to avoid collapse” might go to methodology section. | Done |
Round 2
Reviewer 1 Report
My comments have been taken into account. I agree to the publication.Author Response
Many thanks for the agreement
Reviewer 3 Report
The authors answered to all the revisions.
Author Response
Many thanks for your agreement
Reviewer 4 Report
The authors have adequately addressed most of the concerns. However, the abstract still has 12 lines of background information.
Author Response
Further to your comment, the abstract has now been modified. See final draft.